# A Novel Vision-Based Towing Angle Estimation for Maritime Towing Operations

**Xiong Zou [1], Wenqiang Zhan [1,\*], Changshi Xiao [1,2,3,4,\*], Chunhui Zhou [1,2,3], Qianqian Chen [1], Tiantian Yang [1] and Xin Liu [4,5]**

[1] School of Navigation, Wuhan University of Technology, Wuhan 430063, China;
zx2000@whut.edu.cn (X.Z.); church_zhou@whut.edu.cn (C.Z.); chenqq@ whut.edu.cn (Q.C.);
tiantianyang@whut.edu.cn (T.Y.)

[2] Hubei Key Laboratory of Inland Shipping Technology, Wuhan 430063, China

[3] National Engineering Research Center for Water Transport Safety, Wuhan 430063, China

[4] Institute of Marine Information Technology, Shandong Jiaotong University, Weihai 250357, China;
axinzaixian@163.com

[5] School of Transportation, Wuhan University of Technology, Wuhan 430063, China

**\*** Correspondence: zwq626197298@whut.edu.cn (W.Z.); cs_xiao@hotmail.com (C.X.)

**Abstract:** The demand for non-powered facility towing is increasing with the development of large-scale offshore projects. It is of great interest for its safe operation to measure the state of the towing process in real time. This paper proposed a computer vision algorithm designed to measure the tug yawing during the towing operation by estimating the towing line angle. The geometrical projection of the towing line from 3D to 2D is described in detail. By fixing the camera at specific locations and simplifying the calculation procedure, the towing line angle in the 3D world can be estimated by the line angle in the image. Firstly, the sea–sky line is detected to estimate the rolling angle of the tug in the captured image. Then, the towing line angle is calculated by an image processing method. At the same time, the estimation of the towing angle is achieved through the captured video data analysis. Finally, field experiments were carried out and the results demonstrated that this method is suitable for real-time calculation of the towing angle during the towing operation.

**Keywords:** non-powered facility; towing angle; computer vision; sea–sky line detection

## 1. Introduction

Recently, the towing operation of non-powered facilities at sea has developed rapidly to promote marine exploration, deep sea farming, and meet other marine engineering needs. Offshore drilling platforms, marine breeding platforms, and other non-powered equipment are huge in size. Besides, their hydrodynamics is quite complex and hard to model precisely, especially under the influence of wind, waves, tides, and other natural environmental factors. These factors pose a great challenge to their safe operation. Therefore, there has been an urgent need for industrial development to ensure the safety and reduce the risk of towing operations at sea [1].

Due to the sea's weather conditions, the tug and the towed facility would swing in the scheduled route during the non-powered facility towing voyage, which is the yawing problem of the towing operation. There are many factors that affect the yawing of the towing system, such as the towing speed, the position of the towing point in the towed facility, the length of the rope, and the loading of the towed facility. The yawing would lead to a change in the towing angle. A large towing angle would make the dragging line have a transverse drag force acting on both the tug and the platform. The transverse moment caused by the drag force would result in a transverse inclination, which would influence the system stability. When the towing angle becomes larger, the transverse drag

moment generated by the drag force will become larger, which would increase the wear and stress concentration of the towing line. In this case, the difficulty of the towing operation would also increase; the towing line might even break, causing serious accidents [2].

With the development of computer vision technology, the camera has become an indispensable sensor device in all walks of life due to its low cost, rich information and high resolution; its application in the marine environment is also increasing, such as for obstacles detection, sea–sky line detection, etc. [3].

The main contribution of this paper is that a computer vision-based method is proposed for the towing angle measurement, which contributes to the safety of the non-powered sea platform towing operation. The geometrical projection model of the towing line from 3D to 2D is proposed in the work. Without additional sensors, the towing line angle in the 3D world can be estimated by the line angle in the image. Moreover, the rolling angle of the tug can be estimated by detecting the sea–sky line angle in the captured image. To deal with the adverse effects of surface waves and spray, a specific filter is designed for the towing line detection.

The remainder of this paper is organized as follows. Section 2 reviews related research in the towing field. Section 3 presents the framework of the method, including the mathematics of the projection from the 3D world to the 2D image, the sea–sky line detection method and the process of the towing angle detection in the image coordinate. Section 4 is the experimental section, which provides the experimental evaluation of the method. Finally, Section 5 gives some conclusions.

## 2. Literature Review

### 2.1. Towing Research

Towing operations involve a wide range of fields. Many researchers have done a lot of extensive and in-depth work in the study of the whole process of towing. As early as 1950, performance simulation and stability analyses were carried out by using a steady forward velocity towing model [4]. Bernitsas and Kekrides developed a model to describe the ship's motion towed by an elastic cable [5]. Subsequently, they established the slow-speed dynamic mathematical models of single-line towing and two-line towing in references [6,7], respectively, considering the environmental excitation of the current, wind and average wave drift force. By simulating four systems with different dynamic characteristics, the conclusions from the local linear analysis and global nonlinear analysis were verified. Fitriadhy et al. proposed a numerical model to analyze the stability of the towed ship, and studied the effects of different speeds and different angles of wind on the ship towing system. Under certain wind conditions, the impact tension on the tow line increased, which decreased the towing system stability [8]. Fang and Ju developed a nonlinear mathematical model that takes into consideration the seakeeping and maneuverability of the ship, as well as the influence of wind; simulated the motion characteristics of ships in random waves; and studied the dynamic stability of the towing system in waves [9]. In [10], a mathematical model of a towing system, including a tug, a towing cable and a towed vessel, was established to study the impact of the loading conditions of a towed vessel on the yawing extent of the towing system. The simulation results showed that the towing speed, cable length and loading conditions had a certain influence on the system yaw. Under the stability of the towing system, the relationship between the yawing angle and cable stress was determined, and the limit values of cable stress of the different yawing angles were calculated, in order to evaluate the safety of the towing operation [11].

However, there is very little research about the measurement of real-time yawing. Based on this, the article uses an image processing method to extract the towing line and estimate the yaw of the tug during the towing process, thus ensuring the safety of the towing system.

### 2.2. Sea–Sky Line Detection

The sea–sky line is an important cue for visual perception in the marine environment. In marine images where the sea–sky line is a region boundary, the accurate detection of the sea–sky line is significantly beneficial to the detection of the vessel rolling angle. Dong et al. [12] used the textural

features based on a gray-level co-occurrence matrix to locate the sea–sky line region. Then a set of sea–sky line candidate points were obtained by an image binarization algorithm, and the sea–sky line was detected by the linear fitting method. Wang et al. [13] proposed a method that first acquires the gradient saliency image, and then adopts the region growth method to get the support region, ultimately combining the spatial characteristics to obtain the sea–sky line. Chiyoon et al. proposed a sea–sky line detection method that combined the edge information of different scale images and adopted a convolutional neural network (CNN) to validate the edge pixels belonging to the horizon, and then the sea–sky line was estimated iteratively by least squares fitting [14,15]. Dai et al. [16] used the image segmentation method to extract the edge pixels and applied the Hough transformation to obtain the sea–sky line. Sun et al. [17] proposed a coarse–fine-stitched method for robust horizon line detection. They first used gradient features to build a line candidate pool, applied a hybrid feature filter to extract the fine horizon lines from the pool, and then used the Random Sample Consensus algorithm to obtain the whole horizon line. Wenqiang et al. [18] introduced the local variation method to minimize the influence of background texture and reinforce the sea–sky line structure and adopted the Random Sample Consensus algorithm to fitting the sea–sky line. Subsequently, they [19] proposed a visual detection method for navigable waters based on neural networks online training. Firstly, different regions of the image were clustered, and labels and confidence values were allocated, and then they were fed into the training grid of the neural networks. Finally, the training grid was utilized to segment the input image again, but with higher precision and robustness.

### 2.3. Salient Feature Detection

In this paper, the rope line in the image would be detected by its statistical saliency features. In order to calculate the saliency map and recognize the salient objects in a given image, many researchers propose a lot of saliency models. According to the use of prior knowledge, current methods are divided into two categories: contrast different models and learning-based models. For the contrast difference models, these methods used the local and global center-surround difference with the low-level feature [20]. In the early local-contrast-based method, image pixel contrast was calculated from an image pyramid based on the color and orientation features for saliency detection. Assuming that the salient object has a well-defined closed boundary, Jiang et al. [21] integrated object shape prior and bottom-up salient stimuli to propose a multi-scale contrast-based method. Cheng et al. [22] proposed a regional contrast method for the saliency extraction based on spatial coherence and global appearance contrast, and verified it in the largest public data set. In [23], a hierarchical framework was introduced to obtain a high-response saliency map, which got important values from three image layers in different scales. Bhattacharya et al. [24] proposed a novel algorithm that decomposed the input video into background and residual videos to detect the motion salient regions within much less time. In [25], a novel saliency detection algorithm was described: Multiscale extrema of the local differences measured in the CIELAB space was firstly used to detect potentially salient regions, and then the saliency map was generated by a Gaussian mixture. Chen et al. [26] detailed a new learning framework for video saliency detection, which makes full use of spatiotemporal consistency to improve the detection accuracy. Instead of direct training on image features, Mai et al. [27] trained Conditional Random Fields by saliency aggregation on saliency maps. In [28], a regressor based on discriminative regional features was trained and the image saliency was predicted by a random forest model. Recently, CNN techniques had good performance in salience detection [29], and a novel CNN framework integrated with low-level features was proposed to detect salient objects for complex images [30]. Luo et al. [31] designed a simplified CNN with the local and global information, and proposed a loss function to penalize errors on the edge, inspired by the Mumford–Shah function. A global Recurrent Localization Network (RLN) was proposed to exploit the contextual cue of the weighted response map for salient object detection [32].

### 3. An Overview of the Framework

The towing line angle estimation in the 3D global coordinate is shown in Figure 1. According to the derivation process of the towing line projection, the towing line angle in the 3D global coordinate

could be estimated by the sea–sky line angle and the towing line angle in the image. Then, the image processing procedure is divided into two parallel steps: the sea–sky line angle calculation and the towing line angle calculation in the image, as shown in Figure 1.

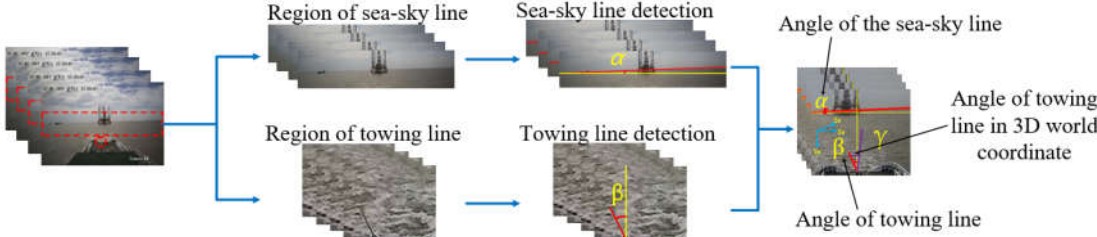

**Figure 1.** The framework of the towing line angle estimation algorithm.

The top diagram shows the process of sea–sky line detection. The vertical gradient map is used for the sea–sky line detection. Before the gradient computation, the image is smoothed by the total variance method to reduce the local texture influence and remain the edge of the background structure. Based on the characteristics of the edge between the sky and the sea, the points with the minimum value are selected as the potential sea–sky line points. Then, the Random Sample Consensus algorithm is adopted to remove the outlier and calculate the sea–sky line angle $\alpha$.

To detect the towing line, the brightness distribution of the image is statistically analyzed, as shown in the bottom diagram. An adaptive threshold is also used to select the potential towing line points. Two new filters are designed to remove the sea wave lines and discrete points. Then the towing line angle in the image is estimated by the least square method. Finally, the towing line angle in the 3D global coordinate is calculated according to two calculated angles, which is also called the towing angle.

### 3.1. The Geometrical Projection Model

The pinhole camera model is regularly employed as a basic image acquisition process; it defines the projection relationship from a 3D global coordinate to a 2D image plane coordinate [33]. The center of the perspective projection is regarded as the optical center. The line that is perpendicular to the image plane and passes through the optical center is regarded as the optical axis. Additionally, the intersection point between the image plane and the optical axis is regarded as the principal point. To simplify the calculation, the camera is mounted to make the optical axis parallel to the sea surface. Simultaneously, the optical axis is in the longitudinal symmetry plane of the observation vessel.

The coordinate system is established, as shown in Figure 2. Origins of world coordinates and the camera coordinates are both at the optical center. The optical axis is set as the $Z_c$ axis and the $Z_w$ axis. In the world coordinate system $P_{1w} = [x_0, y_0, z_0]^T$ is assumed to be the start points of the towing line attached in the tug stern. $P_{2w} = [x_{2w}, y_{2w}, z_{2w}]^T$ is the endpoints of the towing line on the sea surface. $y_{2w}$ is set to be $y_0$ by ignoring the altitude difference of the points $P_{1w}$ and $P_{2w}$. $d$ is the distance of $P_{1w}$ and $P_{2w}$. The angle between the towing line and the $Z_w$ axis in the world coordinate system is towing angle $\gamma$. Then point $P_{2w}$ can be expressed as follows:

$$P_{2w} = \begin{bmatrix} x_{2w} \\ y_{2w} \\ z_{2w} \end{bmatrix} = \begin{bmatrix} x_0 + d\sin\gamma \\ y_0 \\ z_0 + d\cos\gamma \end{bmatrix}. \tag{1}$$

As the pitch value of the tug is small during the towing process, only the rolling transformation is considered from the world coordinate system to the camera coordinate system. The rolling angle coincides with the sea–sky line angle $\alpha$. The transformation matrix $R$ is

$$R = \begin{bmatrix} \cos\alpha & -\sin\alpha & 0 \\ \sin\alpha & \cos\alpha & 0 \\ 0 & 0 & 1 \end{bmatrix}. \tag{2}$$

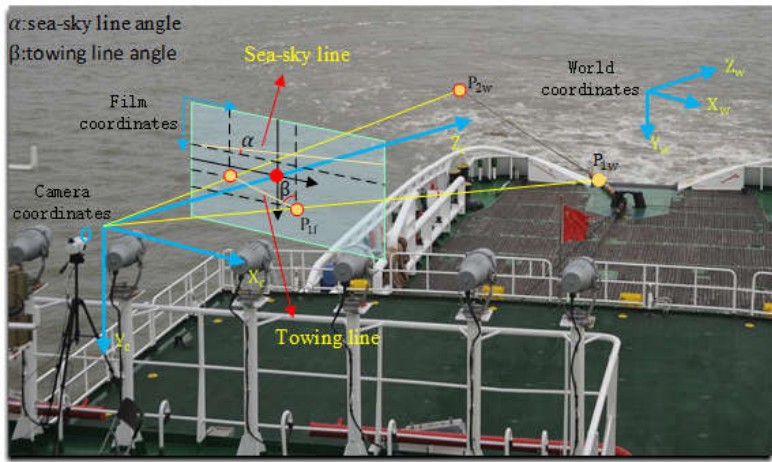

**Figure 2.** The towing line imaging geometry model.

The two towing line points transformed in the camera coordinates are

$$P_{1c} = RP_{1w} = \begin{vmatrix} x_0 cos\alpha - y_0 sin\alpha \\ x_0 sin\alpha + y_0 cos\alpha \\ z_0 \end{vmatrix}, \tag{3}$$

$$P_{2c} = RP_{2w} = \begin{vmatrix} (x_0 + dsin\gamma)cos\alpha - y_0 sin\alpha \\ (x_0 + dsin\gamma)sin\alpha + y_0 cos\alpha \\ z_0 + dcos\gamma \end{vmatrix}. \tag{4}$$

The camera intrinsic matrix is

$$K = \begin{bmatrix} f & 0 & u_0 \\ 0 & f & v_0 \\ 0 & 0 & 1 \end{bmatrix}, \tag{5}$$

where $u_0$ and $v_0$ are the offsets of the horizontal and vertical axes. $P_{1f} = [x_{1f}, y_{1f}]^T$ and $P_{2f} = [x_{2f}, y_{2f}]^T$ are the projected points in the 2D image film, respectively. Besides, the two perspective points of the towing line points in the image pixel coordinates are

$$P_{1f} = KP_{1c} = \begin{bmatrix} \frac{f}{z_0}(x_0 cos\alpha - y_0 sin\alpha) - u_0 \\ \frac{f}{z_0}(x_0 sin\alpha + y_0 cos\alpha) - v_0 \end{bmatrix}, \tag{6}$$

$$P_{2f} = KP_{2c} = \begin{bmatrix} \frac{f}{z_0 + dcos\gamma}((x_0 + dsin\gamma)cos\alpha - y_0 sin\alpha) - u_0 \\ \frac{f}{z_0 + dcos\gamma}((x_0 + dsin\gamma)sin\alpha + y_0 cos\alpha) - v_0 \end{bmatrix}. \tag{7}$$

The towing line angle $\beta$ in the pixel coordinates is estimated by

$$tan\beta = -\frac{x_{2f} - x_{1f}}{y_{2f} - y_{1f}}. \tag{8}$$

As the length of d has no influence on the value of $\beta$, d can be set to infinity. Then, $P_{2f}$ is

$$P_{2f} = \begin{bmatrix} ftan\gamma \\ tan\gamma tan\alpha \end{bmatrix}. \tag{9}$$

According to the formula, we can obtain

$$tan\beta = -\frac{x_0 - y_0 tan\alpha - z_0 tan\gamma}{x_0 tan\alpha + y_0 - z_0 tan\gamma tan\alpha}. \tag{10}$$

For the towing stability, the towing angle is also in the symmetry axis at the stern. $P_{1f}$ is approximately in the middle of the stem of the vessel. Then, $x_0$ is set zero. Then we can get

$$\tan\gamma = \frac{y_0 \tan\alpha - y_0 \tan\beta}{z_0 + z_0 \tan\beta \tan\alpha}. \tag{11}$$

According to the formula, the towing line angle in the 3D global coordinate can be estimated with the parameters $y_0$, $z_0$, $\alpha$ and $\beta$.

### 3.2. Sea–Sky Line Detection

Many observations show that the sea–sky line ranges in the limited area. To reduce the computing expense as well as the noise, the processing area is constrained in the specific region in the image that contains the sea–sky line.

The sea–sky line is detected based on the max change at the edge of the sea and the sky. The points with max gradient value in each column are regarded as the sea–sky line points. As the textures of the cloud and the sea wave are featured with great variation, points in these regions have a large chance to be selected as the potential points. To solve this problem, the sky and water regions should be smoothed and a max change in the sea–sky line region should remain at the same time.

The traditional denoising algorithms are likely to smooth both the local texture and the global structure. To address this problem, the total variation algorithm proposed by Xu is applied [34].

The sky region above the sea–sky line is lighter. The area below the sea–sky line is composed of darker areas like the sea and ships. The points along the vertical lines with an absolute great gradient value are possible sea–sky line points. As the y-axis of the image coordinate system is from top to bottom, the gradient value of the sea–sky line points is negative. Then, points with the minimum value in each column of the gradient map are selected as the potential sea–sky line points.

The selected sea–sky line points are used to estimate the line parameter. Even though the gray image is smooth in the local texture, we could not ensure that the minimum vertical gradient value appears in the sea–sky line. To reduce the outlier point influence, Random Sample Consensus (RANSAC) was adopted to estimate the sea–sky line angle. The algorithm can obtain the correct parameter values under a large amount of noise through iterative calculation [35].

### 3.3. Towing Line Angle Estimation in Image Pixel Coordination

In this section, the process of the towing line detection and the angle estimation in the towing vessel is described. A new saliency selection method is used to detect the towing line. Then a novel filter method is applied to remove the sea wave texture and the stern spray noise points. After that, the towing line angle is estimated by the towing line points.

### 3.3.1. Towing Line Selection

As the camera is mounted on the stern of the towing vessel, the towing line appears in the limited region in the video. Thus, only the interest region is processed to detect the towing line. Intuitively, the towing line is outstanding from the sea surface background. Inspired by saliency detection, which is described as an attention mechanism in organisms to narrow down to the remarkable parts of what they see, the statistical and geometric linear characteristic is adapted to detect the towing line from the complex background.

The color of the towing line is black in the video. Considering the influence of the sea wave, it is difficult to detect the towing based on the probability distribution of the brightness in the image, as shown in Figure 3. Intuitively, the towing line region is dark in the image, and the value of the towing line accounts for a small part in the brightness statistics. Then the brightness distribution of the towing line region is considered as the Gaussian distribution.

$$P(x,y) = \frac{1}{\sigma\sqrt{2\pi}} e^{-(I(x,y)-\mu)^2/2\sigma^2}. \tag{12}$$

The points with high-density probability are background, while the points with low-density probability have a great chance to be towing line points.

$$B(x,y) = \begin{cases} 0 & if \quad P(x,y) > P_T \\ 1 & otherwise \end{cases} , \tag{13}$$

where $B(x,y)$ is the background scene model, and $P_T$ is the threshold. Then, the mean and standard deviation of the towing line region is calculated as

$$\mu = \frac{1}{H \times W} \sum_{x-1}^{W} \sum_{y-1}^{H} I(x,y), \tag{14}$$

$$\sigma = \sqrt{\frac{1}{H \times W - 1} \sum_{x-1}^{W} \sum_{y-1}^{H} (I(x,y) - \mu)^2}. \tag{15}$$

Considering the towing line brightness with a low value, the brightness value of the towing line is on the left side of the brightness distribution. Then, the towing line brightness value should satisfy

$$\frac{\mu - I(x,y)}{\sigma} > T, \tag{16}$$

where T is the threshold value. After several experiments, it was found that a good result can be obtained when T is 2.

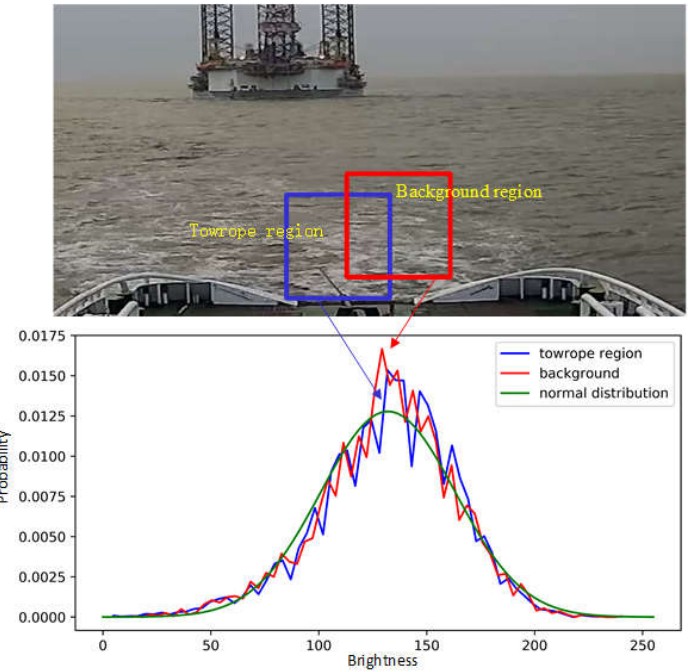

**Figure 3.** Probability distribution of the brightness.

### 3.3.2. Filtering Horizontal Wave Line Points

After selecting the potential towing line points with the statistical method, there is still a lot of noise in the selected points. The sea wave results in the noise points that are displayed as a horizon line in the image. In addition to that, the stern spray is presented as the discrete noise points. The potential towing line points are modeled and filtered by adopting the statistical method as three partitions:

$$P = P_t \cup P_h \cup P_d, \tag{17}$$

where P is the set of potential towing line points; $P_t$ is the set of towing line points; $P_h$ is the set of sea wave horizontal line points; and $P_d$ is the set of discrete noise points.

To remove the noise and get the towline points, first of all, the horizontal wave line points are removed. Then, based on the line property, the discrete points are denoised. Rather than removing the horizontal line points directly, a filter is used to retain the horizontal line point $P_h$, as shown in

Figure 4c. Then, the potential towing line point set P is subtracted from the horizontal line point set $P_h$ to get the towing line points and the discrete noise point set $P_{td}$.

$$P_{td} = P - P_h. \tag{18}$$

Morphology is applied to detect continuous horizon line points. The horizontal filter is shown in Figure 4a. The potential towing line point map is convoluted with the horizontal filter. Map points of the value equal to three are set to be one, while the others are set to be zero. When one of the horizontal neighbors is not in the potential towing line point set, the potential points are removed from the set. The discrete points and towing line points would be removed, with the horizontal sea wave line points remaining. After the erosion operation, the eroded potential point map is dilated by convoluting with the horizontal filter again. At this time, points with a non-zero value are set to be one. The dilating is to offset the erosion of the horizontal wave line points.

$$M_h = D(E(M * F_H) * F_H), \tag{19}$$

where $F_H$ is the horizontal filter kernel; M is the potential towing line point map; $M_h$ is the horizontal wave line point map; and $E(\cdot)$ and $D(\cdot)$ are relevant operations of erosion and dilation respectively.

$$E(x,y) = \begin{cases} 1 & if \quad I(x,y) > 3 \\ 0 & otherwise \end{cases}. \tag{20}$$

### 3.3.3. Filtering Discrete Points

After removing the horizontal wave line points, the final step is to remove the discrete noise points. The traditional method of removing small objects from the foreground is the morphology operation of opening, which is the combination of the erosion operation and dilation operation. In the opening operation, the erosion operation removes the small objects, and the dilation operation restores the shape of the remaining objects. As the erosion operation erodes objects in all directions, line points could be removed at the same time. To solve this problem, a novel method was designed to remove the discrete points and keep the line points. The new method consists of three convolution kernels, as shown in Figure 5.

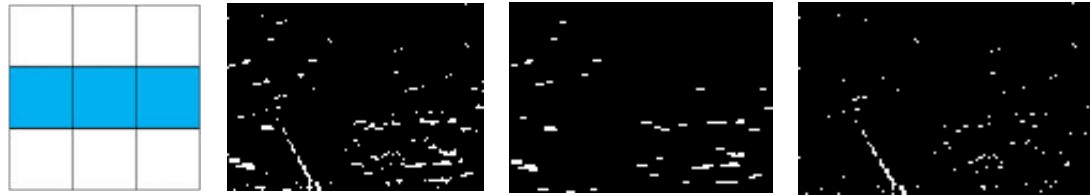

**Figure 4.** Horizontal wave line points removing: (**a**) the horizontal filter; (**b**) the potential towing line points map; (**c**) the horizontal line points filtered by the horizontal filter; (**d**) the final result map.

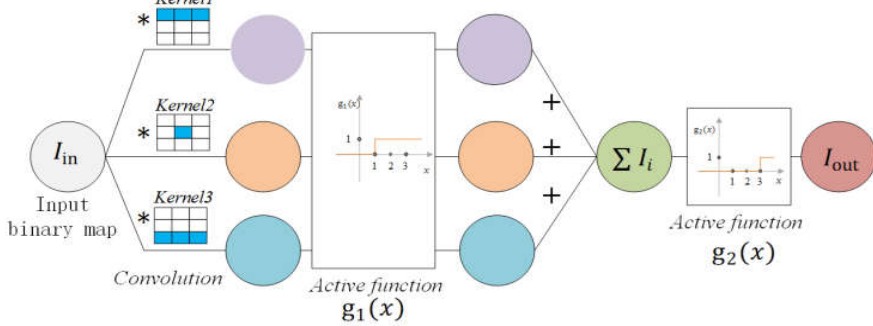

**Figure 5.** The discrete points filter.

The point map is convoluted by the three kernels and processed through the active function $g_1(x)$, respectively. Then these three convoluted maps are summed up. Finally, the active function $g_2(x)$ is used to filter the discrete points.

$$g_1(x) = \begin{cases} 1 & if \quad x > 1 \\ 0 & otherwise \end{cases}, \tag{21}$$

$$g_2(x) = \begin{cases} 1 & if \quad x = 3 \\ 0 & otherwise \end{cases}. \tag{22}$$

Kernel 2 is used to detect whether each location is featured with points. Kernels 1 and 3 are to detect whether the up and the down locations have points. The active function $g_1(x)$ is the binary convoluted map. The active function $g_2(x)$ tends to detect the up and down neighbor of the points at the same time. Then the points are removed when either their up or down points do not exist.

Figure 6 shows the effect of the filtering method. With the vertical and oblique continuity, it is supposed that there is more than one point at each of the upside and downside of the towing line points. The filtering method is to remove the discrete points and keep the towing line points, as shown in Figure 6b.

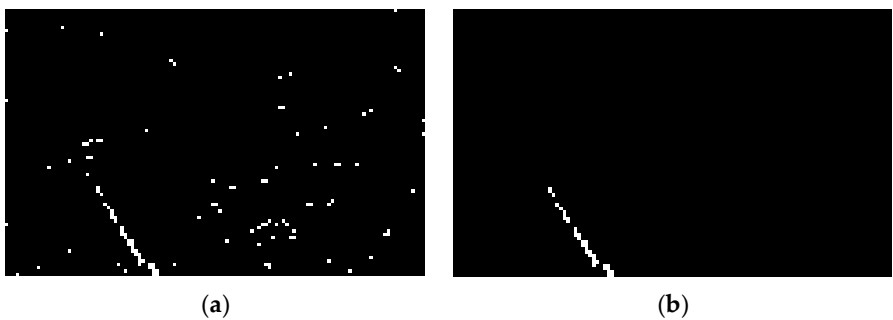

(a)          (b)

**Figure 6.** Discrete points removal process: (**a**) map with the discrete points; (**b**) the final towing line.

### 3.3.4. Line Parameter Calculation

The least-square method is adopted to estimate the towing line parameter. To avoid the towing line vertical to the horizontal line, we model the towing line as

$$x = k \cdot y + b. \tag{23}$$

The objective is to adjust the parameters of the line model function to best fit the points, $(x_i, y_i), i = 1, \ldots, n$. The least-squares method can be used to find its optimum with min squared sum residuals:

$$\arg\min_{k,b} \sum_i (x_i - k \cdot y_i - b)^2. \tag{24}$$

The min squared sum residuals are found when the gradient is zero. Then we can derive the values of k and b that minimize the objective function:

$$\hat{k} = \frac{\sum_i (x_i - \bar{x})(y_i - \bar{y})}{\sum_i (y_i - \bar{y})^2}, \tag{25}$$

where $\bar{x}$ and $\bar{y}$ are the averages of $x_i$ and $y_i$, respectively. Then, the angle of the sea–sky line is estimated by

$$\alpha = \tan K. \tag{26}$$

## 4. Experiments and Discussion

Several experiments were conducted to examine the validity of our method. Our experiment video data was recorded when the vessel towed the jack-up drilling platform. As some parameters are internal, secret information, this parameter is estimated roughly. The experiments successfully estimated the towing line angle and the result of the angle was consistent with empirical values.

### 4.1. Experimental Setup

Our experimental data relied on a towing operation, when the vessel DONG HAI JIU 101 towed the CJ50 jack-up drilling platform H1418 in 24 h. The H1418 platform features a square structure that has a plane size of 70 m × 68 m. The height is about 82 m and the total weight is about 17,100 tons. These parameters were roughly estimated due to internal confidentiality. The vision system is placed at the tail of the vessel and consists of an industrial camera (200 W pixel, 3.2 μm pixel size) and industrial lens (12 mm, 1:1.4), and the captured images were recorded at 25 f/s, with a resolution of 1920 × 1080 pixels.

### 4.2. Water Line Detection

Figure 7 shows the results of the sea–sky line detection in different light conditions. The second column in Figure 7 is the point map with the maximum value in each column of the vertical gradient map. The third column is the result of the sea–sky line using the RANSAC algorithm based on the second column. As can be seen from the second column, due to the interference of various noises, the points of the maximum gradient value are not completely collinear, but most values are collinear. The RANSAC algorithm is very robust to outliers, and can estimate the required parameters well in the case of a large number of outliers. As shown in the third column, the correct sea–sky line results can be obtained within a few dozen iterations.

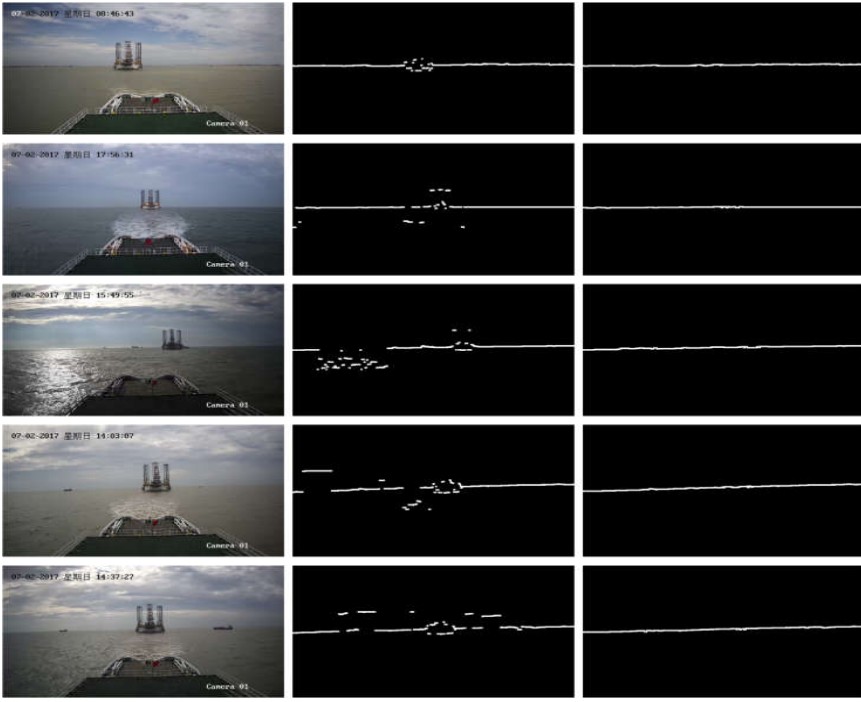

**Figure 7.** The sea–sky line detection process: the **first** column is the original image; the **second** column is the points with the maximum gradient value in each column of the image; and the **third** column is the sea–sky line detection by the RANSAC algorithm.

### 4.3. Towing Line Detection

Several saliency detection methods were adopted to locate the towing line in the image. Figure 8 shows the result of the context-aware-based saliency detection methods (CA) [36]; dense and sparse reconstruction-based saliency detection methods (DSR) [37]; Markov chain-based saliency detection methods (MC) [38]; spectral residual-based saliency detection methods (SR) [39]; and Bayesian-based statistics saliency detection methods (SUN) [40]. The context-aware-based and spectral residual-based methods could roughly locate the towing line. However, this method is susceptible to sea wave

texture. The dense and sparse reconstruction-based method and the Markov chain-based method failed to detect salient line objects. The Bayesian-based method was influenced by the white sea spray. All these saliency detection methods could not satisfy our requirement to detect the towing line in the image.

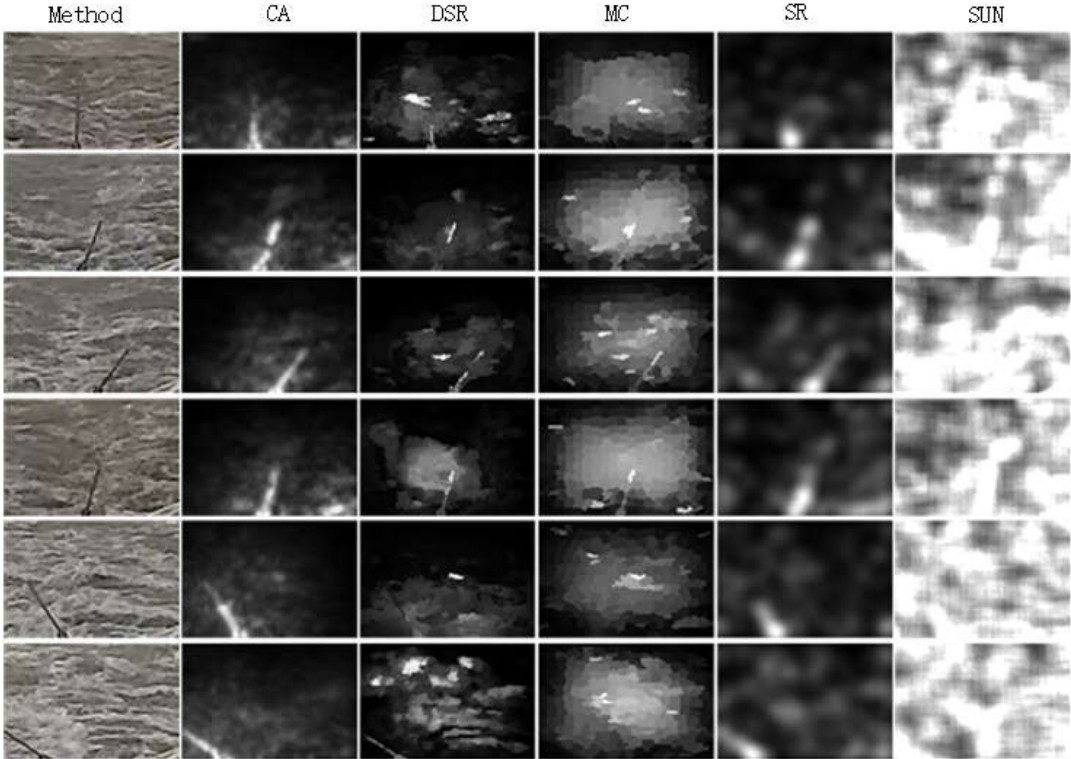

**Figure 8.** Different saliency detection of the towing line: the **first** column is the original image; the **second** column is the result by CA methods; the **third** column is the result by DSR methods; the **fourth** column is the result by MC methods; the **fifth** column is the result by SR methods; and the **sixth** column is the result by SUN methods.

Canny edge detection and Hough methods were adopted to detect the towing line, as shown in Figure 9. The second column of the figure is the result of Canny edge detection. Most of the edges are the sea wave. The fourth column is the result of the Hough method after Canny edge detection. However, this method could not detect the towing line. The third column is the potential towing line point map by using our salient statistical method. As there are many noise points in these points, the Hough method could not exactly detect the towing line, as shown in the fifth column. Experiments show that our method could effectively detect the towing line points shown in the last column.

Figure 10 is the result of our method. The second column is the potential towing line points. The third column is the detection of horizontal sea wave line points. By subtracting the horizontal line points from the potential towing line points, the remaining discrete and towing line point map is shown in the fourth column. The last column is the final towing line detection. Through the previous comparative experiment and the step-by-step result analysis of the proposed algorithm, it demonstrates that the proposed method performs well, regardless of the complex sea surface with waves and spray.

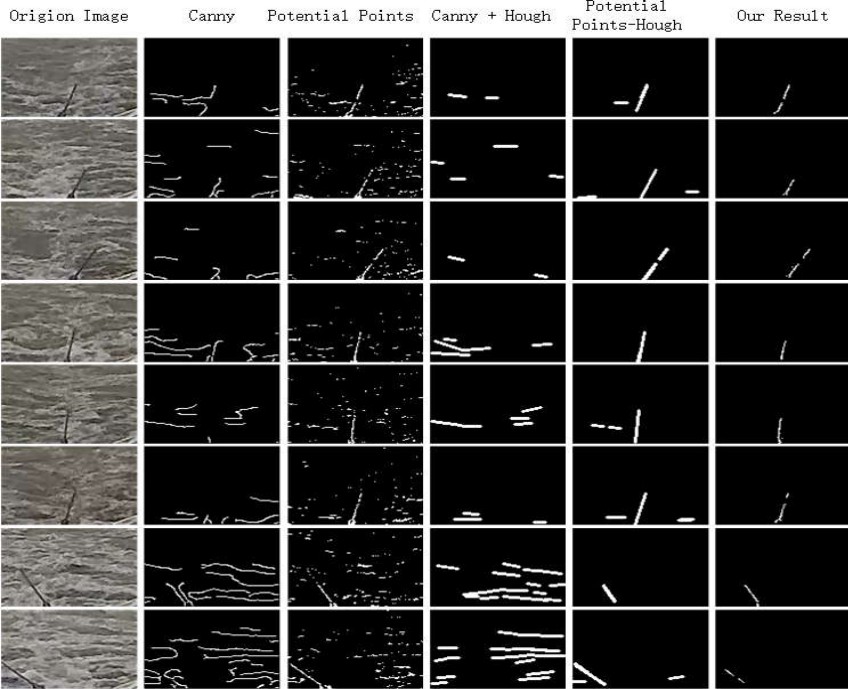

**Figure 9.** Experiment comparison with other method: the **first** column is the original image; the **second** column is Canny edge detection; the **third** column is the potential towing line point detection by our salient statistical method; the **fourth** column is Hough refinement after Canny edge detection; the **fifth** column is Hough refinement after our potential towing line point detection method; and the **sixth** column is the proposed methods.

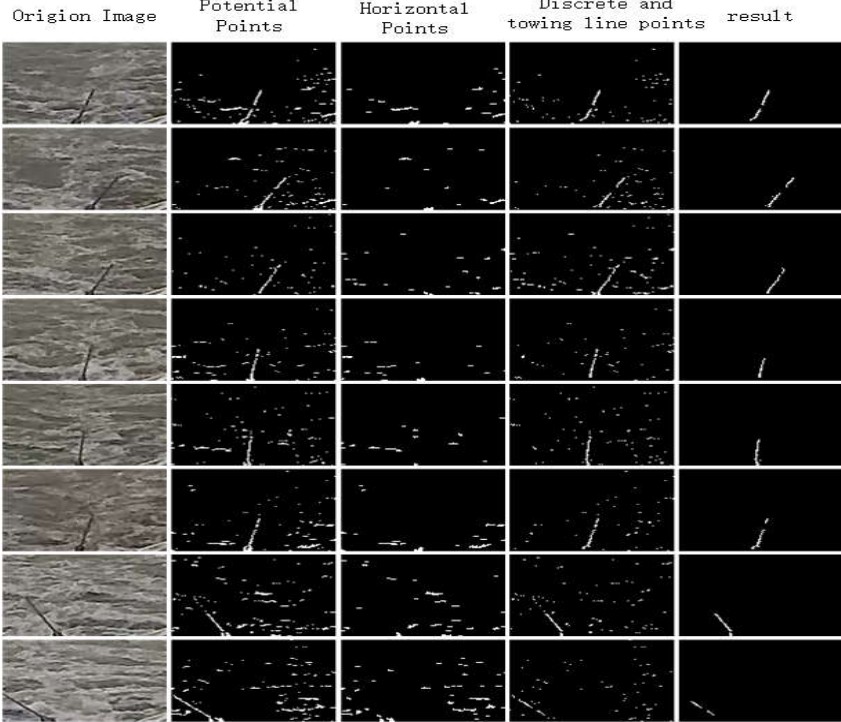

**Figure 10.** Our towing line detection method: the **first** column is the original image; the **second** column is the potential towing line point detection; the **third** column is horizontal sea wave line point

detection; the **fourth** column is the remaining points by subtracting the horizontal line points from the potential towing line points; and the **fifth** column is the final towing line detection.

### 4.4. Towing Line Angle Estimation

The information of the vessel is illustrated in Table 1.

**Table 1.** The vessel parameters.

| Ship Type | Built | Draught | Length | Width | Gross Tonnage |
|---|---|---|---|---|---|
| Search and Rescue Vessel | 2012 | 6 m | 117 m | 16 m | 4747 t |

The camera was mounted about 10 m above the sea surface and 70 m away from the stem of the vessel; that is to say, $y_0 = 10$ and $z_0 = 70$. According to Equation (11), the towing line angle in the 3D world coordination could be estimated by

$$\tan \gamma = \frac{10 \tan\alpha - 10 \tan\beta}{70 + 70 \tan\beta\tan\alpha}. \tag{27}$$

The following figure is the result of the angle detection in continuous time. The angles are measured in degrees. The rolling angle of the towing vessel could be estimated by the sea–sky line detection. When there is no rolling in the towing vessel, the sea–sky line angle in the image should be zero. Figure 11 shows that the vessel rolling angle in the image ranges from −2.53° to −0.62° degrees. The rolling angle does not fluctuate near zero because of the bias of the vessel heels. According to the Figure, the navigation state of the towing vessel can be obtained. The frequency of the vessel rolling could also be calculated. Figure 12 shows the towing line angle value at different times in the image. Based on this information, the towing line angle value in the 3D global coordination can be estimated according to Equation (27).

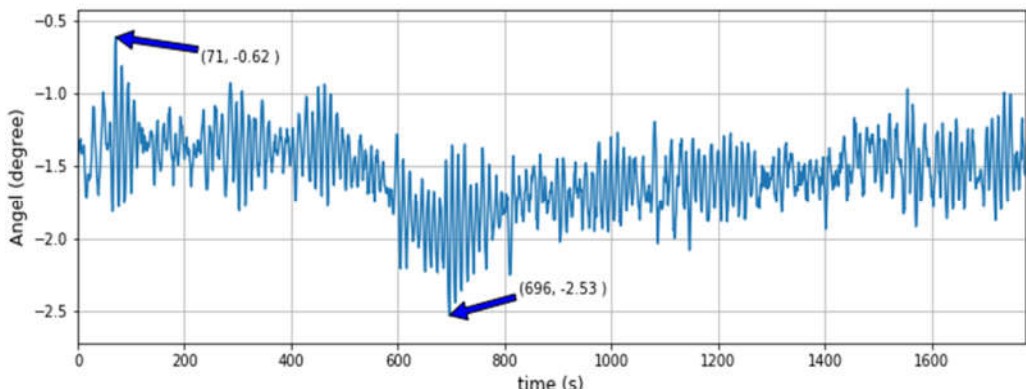

**Figure 11.** The rolling angle of the towing vessel.

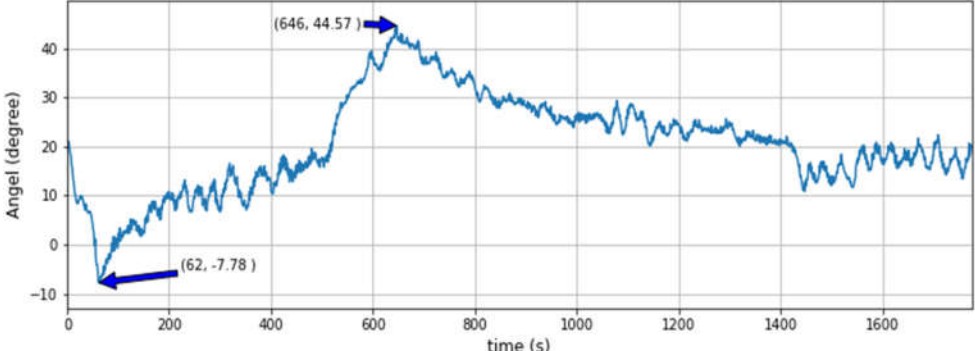

**Figure 12.** The towing line angle in 2D pixel coordination.

To verify the effectiveness of the proposed method, we compared the towing line angle prediction based on the image with the prediction based on the GPS information. Two GPS devices were placed on the towing vessel and towed platform, respectively, to record their trajectory. The towing line was approximately regarded as the line $\overrightarrow{P_1P_2}$ between the towing vessel and towed platform, as shown in Figure 13. An electronic compass was used to detect the course of the towing vessel $\vec{v}$, Then, the angle $\theta$ between $\overrightarrow{P_1P_2}$ and $\vec{v}$ was approximatively regarded as the contrast value of the towing line angle.

Figure 14 shows the towing angles predicted by our proposed method based on the vision information and the compared prediction based on the GPS. According to our proposed method, the estimated 3D global towing line angle ranges from $-1.00°$ to $7.58°$ degrees, which is consistent with the GPS prediction.

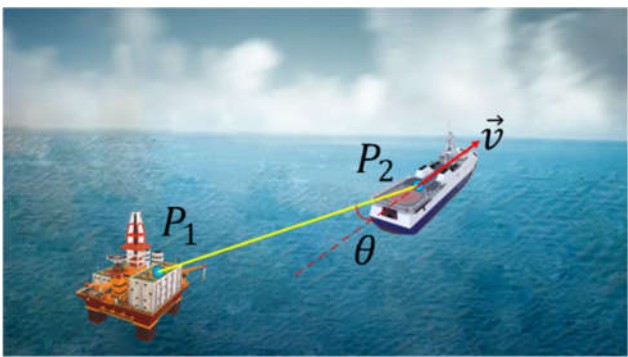

**Figure 13.** The towing line angle in 3D global coordination.

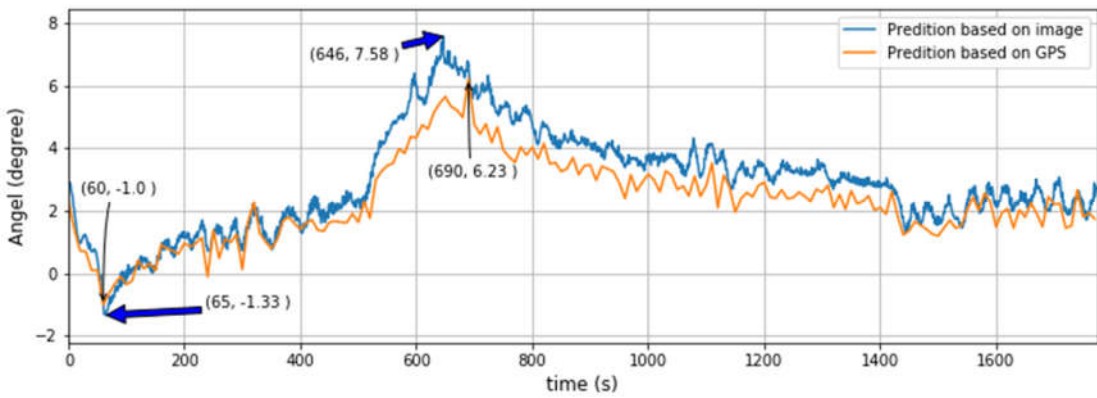

**Figure 14.** The towing line angle in 3D global coordination.

## 5. Conclusions

In this paper, a computer vision-based method is proposed to estimate the towing line angle, which contributes to the safety of the non-powered sea platform towing operation. The towing line angle in the 3D world can be estimated with the captured image according to the proposed geometrical projection model of the towing line from 3D to 2D. Meanwhile, the rolling angle of the tug can also be estimated by detecting the sea–sky line. Our proposed method is robust to deal with the effects of surface waves and spray, as the new method is designed to detect the towing line and filter out the sea wave line and stern spray points. Through the experiments, the validity of the method was verified, achieving the towing angle real-time calculation during the towing operation.

Therefore, our method can be potentially used to ensure the safety of the towing system. However, there is still much room for improvement in the future; for example, the influence of the

height and distance of the tug and the towed platform on the towing line angle will affect the calculation of the towing angle, which is our next research aim.

**Author Contributions:** X.Z. and W.Z. conceived and designed the experiments; X.Z., W.Z. and X.L. prepared the materials and performed the experiments; C.X. and W.Z. contributed the experimental platform and the experimental materials; X.Z. and C.Z. analyzed the data; X.Z. and W.Z. wrote the paper; Q.C. and T.Y. revised the paper. All authors have read and agreed to the published version of the manuscript.

**Funding:** This research was funded by the National Key R & D Program of China (No. 2018YFC1407405, 2018YFC0213904); the National Science Foundation of China (NSFC) (No. 51579204, 51679180, 41801375, 51709218); Wuhan University of Technology Independent Innovation Research Foundation of China through Grant No. 2018III059GX, Hubei Provincial Natural Science Foundation of China through Grant No. 2016CFB362, and the Double First-Rate Project of WUT.

**Acknowledgments:** The authors would like to thank all their colleagues in this project for their help.

**Conflicts of Interest:** The authors declare no conflict of interest.

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
