# Peer review of "A Novel Vision-Based Towing Angle Estimation for Maritime Towing Operations"

_jmse, doi:10.3390/jmse8050356_

Round 1

Reviewer 1 Report

please see the attached document.

Author Response

Response to Reviewer 1 Comments

Point 1: in paragraph 3, the introduction of the used computer vision-based technique is missing.

Response 1: Thanks. This is a very suggestion.

The introduction of the used computer vision-based technique is reallyl lacking, so we made the following modification.

With the development of computer vision technology, the camera has become an indispensable sensor device in all walks of life due to its low cost, rich information and high resolution. And its application in the marine environment is also increasing, such as the obstacles detection, sea-sky line detection, etc. [3]

Point 2:

1) In section 2.1 Towing Research, please add some research work regarding the mathematical study of the towing system.

2) In section 2.2 Sea-sky Line Detection, please mention the full name of abbreviations when its used in the first time, for example, OSTU algorithm.

3) In section 2.3 Salient Feature Detection, from line 125 to 136, it is really confusing.

4) the advantage of the proposed algorithm should be emphasized.

Response 2: Thanks. We appreciate your thoughtful comment.

1).The section 2.1 was modified as follow:

Towing operation involves a wide range of fields. Many researchers have done a lot of extensive and in-depth work in the study of the whole process of towing. As early as 1950, it carried out performance simulation and stability analysis using a steady forward velocity towing model [4]. Bernitsas and Kekrides developed a model to describe the ship motion towed by an elastic cable [5]. Subsequently, they established the slow-speed dynamic mathematical models of single-line towing and two-line towing in the references [6] and [7], respectively, taking into account the environmental excitation of current, wind and average wave drift force. By simulating four systems with different dynamic characteristics, the conclusions of local linear analysis and global nonlinear analysis are verified. Fitriadhy et al. proposed a numerical model to analyze the stability of the towed ship, and studied the effects of different speeds and different angles of wind on the ship towing system. Under certain wind conditions, the impact tension on the tow line increased, leading to decrease the towing system stability [8]. Fang and Ju developed a nonlinear mathematical model that takes into consideration the seakeeping and maneuverability of the ship as well as the influence of wind, simulated the motion characteristics of ships in random waves, and studied the dynamic stability of the towing system in waves [9]. In [10], a mathematical model of a towing system including a tug, a towing cable, and a towed vessel was established to study the impact of loading conditions of a towed vessel on the yawing extent of the towing system, and the simulation results show that the towing speed, cable length and loading conditions have certain influence on the system yaw. Under the stability of the towing system, the relationship between yawing angle and cable stress was determined, and the limit values of cable stress of different yawing angles were calculated, in order to evaluate the safety of the towing operation [11].

However, there are very little researches about the measurement of real-time yawing. Based on this, the article uses image processing method to extract the towing line and estimate the yaw of the tug during the towing process, thus ensuring the safety of the towing system.

2). In section 2.2 Sea-sky Line Detection, please mention the full name of abbreviations when its used in the first time, for example, OSTU algorithm.

OSTU is a person's name, and it was modified as follow:

Line 80, the sentence "... by the OTSU algorithm ..." was modified to "... by image binarization algorithm..."

The section 2.2 was modified as follow:

The sea-sky line is an important cue for visual perception in the marine environment. In marine images where the sea-sky line is a region boundary, the accurate detection of the sea-sky line is significantly beneficial to the detection of the vessel rolling angle. Dong et al. [12] used the textural features based on gray-level co-occurrence matrix to locate the sea-sky line region. Then a set of sea-sky line candidate points are obtained by image binarization algorithm, and the sea-sky is detected by the linear fitting method. Wang et al. [13] proposed the method that first acquire gradient saliency image, then adopt the region growth method to get support region, and combine the spatial characteristics to obtain sea-sky line. Chiyoon et al. proposed a sea-sky line detection method which combined the edge information of different scale images and adopted CNN to validate edge pixels belonging to the horizon, and then the sea-sky line was estimated iteratively by least squares fitting [14][15]. Dai et al. [16] used image segmentation method to extract the edge pixels and applied Hough transform to obtain the sea-sky line. Sun et al. [17] proposed a coarse-fine-stitched method for the robust horizon line detection. They first used gradient features to build a line candidate pool, and applied a hybrid feature filter to extract fine horizon lines from the pool, and then used Random Sample Consensus algorithm to obtain the whole horizon line. Wenqiang et al. [18] introduced the local variation method to minimize the influence of background texture and reinforce the sea-sky line structure and adopted Random Sample Consensus algorithm to fitting the sea-sky line. Subsequently, they [19] proposed a visual detection method for navigable waters based on neural networks online training. Firstly, different regions of the image were clustered, and labels and confidence values were allocated, and then they were fed into the training grid of neural networks. Finally, the training grid was utilized to segment the input image again, but with higher precision and robustness.

3). In section 2.3 Salient Feature Detection, from line 125 to 136, it is really confusing.

They have been deleted

The section 2.3 was modified as follow:

In this paper, the rope line in the image would be detected by its statistical saliency features. In order to calculate the saliency map and recognize the salient objects in a given image, many researchers propose a lot of saliency models. According to the use of prior knowledge, current methods are divided into two categories: contrast different models and learning-based models. For the contrast difference models, these methods used the local and global center-surround difference with the low-level feature [20]. In the early local-contrast based method, image pixel contrast was calculated from an image pyramid based on color and orientation features for saliency detection. Assuming that the salient object has a well-defined closed boundary, Jiang et al. [21] integrated object shape prior and bottom-up salient stimuli to propose a multi-scale contrast-based method. Cheng et al. [22] proposed a regional contrast method for the saliency extraction based on spatial coherence and global appearance contrast, and verified it in the largest public data set. In [23], a hierarchical framework was introduced to obtain high response saliency map, which got important values from three image layers in different scales. Bhattacharya et al. [24] proposed a novel algorithm that decomposed the input video into background and residual videos to detect the motion salient regions with much lesser time. In [25], a novel saliency detection algorithm was described, multiscale extrema of local differences measured in the CIELAB space was firstly used to detect potentially salient regions, and then the saliency map was generated by a Gaussian mixture. Chen et al. [26] detailed a new learning framework for video saliency detection, which maked full use of the spatialtemporal consistency to improve the detection accuracy. Instead of direct training on image features, Mai et al. [27] trained a Conditional Random Fields by saliency aggregation on saliency maps. In [28], a regressor based on discriminative regional features is trained and the image saliency is predicted by a random forest model. Recently, convolutional neural network (CNN) techniques had good performance in salience detection [29], and a novel CNN framework integrated with low-level features was proposed to detect salient objects for complex images [30]. Luo et al. [31] designed a simplified CNN with the local and global information, and proposed a loss function to penalize errors on the edge, inspired by the MumfordShah function. A global Recurrent Localization Network (RLN) was proposed to exploit contextual cue of the weighted response map for salient object detection [32].

4). the advantage of the proposed algorithm should be emphasized.

Line 49, the second last paragraph of the Introduction was modified as follow:

The main contribution of this paper is that a computer vision-based method is proposed for the towing angle measurement, which contributes to the safety of the non-powered sea platform towing operation. The geometrical projection model of the towing line from 3D to 2D is proposed in the work. Without additional sensors, the towing line angle in the 3D world can be estimated by the line angle in the image. Moreover, the rolling angle of the tug can be estimated by detecting the sea-sky line angle in the captured image. To deal with the effects of surface waves and spray, a specific filter is designed for the towing line detection.

Point 3:  An Overview of the Framework:

1) Please enlarge figure 1. It is hard to read.

2) The corresponding literature of the geometrical projection model should be cited

3) Please check eq. (5), the definition of f, uo and vo is not defined.

4) Please check eq. (10), what is a?

5) The used method RANSAC is not introduced or cited.

Response 3: Thanks. We have made the following modification:

1). Please enlarge figure 1. It is hard to read.

The figure 1 is as follow:

2). The corresponding literature of the geometrical projection model should be cited

Line 160, the sentence “The pinhole camera model is regularly employed as a basic image acquisition process, it defines the projection relationship from a 3D global coordinate to a 2D image plane coordinate [33]”

3). Please check eq. (5), the definition of f, uo and vo is not defined.

Line 182, the sentence "Where and  are the offsets of the horizontal and vertical axes." is added.

4). Please check eq. (10), what is a?

Line 178, the sentence "The rolling angle α could be estimated from the water line angle."

5). The used method RANSAC is not introduced or cited.

Line 209, the sentence "To reduce the outlier point influence, we use the Random Sample Consensus (RANSAC) to estimate the sea-sky line angle." was modified to " To reduce the outlier point influence, the Random Sample Consensus (RANSAC) is adopted to estimate the sea-sky line angle. The algorithm can obtain the correct parameter values under a large amount of noise through iterative calculation [35]."

Point 4: Experiments and Discussion:

1) please consider adding an extra section for experimental setup

2) in section 4.1 Water Line Dection, please add more description of figure 7 and.please consider comparing the results with other methods.

3) please clarify the advantage of the used method.

Response 4: Thanks. We agree with the reviewer.

1). please consider adding an extra section for experimental setup

section4.1 is added.

4.1. Experimental Setup

Our experiment data relies on the towing operation, when the vessel of DONG HAI JIU 101 towed the CJ50 jack-up drilling platform H1418 in 24 hours. The H1418 platform is featured with a square structure that has a plane size of 70 meters × 68 meters. The height is about 82 meters and the total weight is about 17100 tons. This parameter are roughly estimated due to internal confidentiality. The vision system is placed at the tail of the vessel and consists of an industrial camera (200W pixel, 3.2 um pixel size) and industrial lens (12mm, 1:1.4), and the captured images were recorded at 25 f/s, with a resolution of 1920 × 1080 pixels.

2). in section 4.1 Water Line Dection, please add more description of figure 7 and.please consider comparing the results with other methods.

Line 315, the sentence "The third column is the detected sea-sky line by the RANSAC algorithm. With the influence of the sun reflection and the object in the sea, the points with the minimum gradient value are not likely to be the sea-sky line points in each column. The RANSAC algorithm is robust to the outliers and can remove this noise points, and the sea-sky line is estimated by the inliers." was modified to "The third column is the result of sea-sky line by the RANSAC algorithm based on the second column. As can be seen from the second column, due to the interference of various noises, the points of the maximum gradient value are not completely collinear, but the most values are collinear. The RANSAC algorithm is very robust to the outliers, and can estimate the required parameters well in the case of a large number of outliers. As shown in the third column, the correct sea-sky line results can be obtained within a few dozen iterations."

For the sea-sky line detection, other methods are not much different from ours in nature. Most methods use RANSAC algorithm for straight line fitting, which is not our main innovation.

3). please clarify the advantage of the used method.

Line 357, add the following sentences: "Through the previous comparative experiment and the step by step result analysis of the proposed algorithm, it demonstrates that the proposed method has a good performance regardless the complex sea surface with waves and spray."

Point 5: Conclusions:

Please emphasize the main contribution part.

Response 4: Thanks. We agree with the reviewer.

The last paragraph of the article was modified as follow:

In this paper, a computer vision-based method is proposed to estimate the towing line angle, which contributes to the safety of the non-powered sea platform towing operation. The towing line angle in the 3D world can be estimated with the captured image according to the proposed geometrical projection model of the towing line from 3D to 2D. Meanwhile, the rolling angle of the tug can also be estimated by detecting the sea-sky line angle. Our proposed method is robust to deal with the effects of surface waves and spray, as a new method is designed to detect the towing line and filter out the sea wave line and stern spray points. By the experiments, the validity of the method is verified to achieve the process of real-time calculation regarding the yawing angle in the 3D global coordinates during the towing operation.

Therefore, our method can be potentially used to ensure the safety of the towing system. However, there is still much room for improvement in the future, for example, the influence of the height and distance of the tug and the towed platform on the towing line angle β will effect the calculation of angle γ, which is the next research content.

Reviewer 2 Report

The main concerns are as follows.

  1. It's regret that the authors ignored the altitude difference of two towing points P_{1w} and P_{2w}. 5~10deg. of towing angle might make significant effect on the calculation of angle \gamma.

Minors are

  1. In subsection 2.3, the last three paragraph should be deleted.
  2. Figure 1 and Figure 2 would be much better to be enlarged or more clearly depicted so as for the readers to easily follow.

Author Response

Response to Reviewer 2 Comments

 Point 1: It's regret that the authors ignored the altitude difference of two towing points P_{1w} and P_{2w}. 5~10deg. of towing angle might make significant effect on the calculation of angle \gamma.

Response 1: Thanks. This is a very suggestion.

The ship floats up and down as the influence the waves during the voyage. And the altitude is changing all the time. What’s more, the value of the altitude is too small compared to the length of the towing line. Therefore, we conducted a preliminary study here, and neglected the influence of the altitude, which is the next research content.

Point 2: In subsection 2.3, the last three paragraph should be deleted.

Response 2: Thanks. We agree with the reviewer.

They have been deleted

The section 2.3 was modified as follow:

In this paper, the rope line in the image would be detected by its statistical saliency features. In order to calculate the saliency map and recognize the salient objects in a given image, many researchers propose a lot of saliency models. According to the use of prior knowledge, current methods are divided into two categories: contrast different models and learning-based models. For the contrast difference models, these methods used the local and global center-surround difference with the low-level feature [20]. In the early local-contrast based method, image pixel contrast was calculated from an image pyramid based on color and orientation features for saliency detection. Assuming that the salient object has a well-defined closed boundary, Jiang et al. [21] integrated object shape prior and bottom-up salient stimuli to propose a multi-scale contrast-based method. Cheng et al. [22] proposed a regional contrast method for the saliency extraction based on spatial coherence and global appearance contrast, and verified it in the largest public data set. In [23], a hierarchical framework was introduced to obtain high response saliency map, which got important values from three image layers in different scales. Bhattacharya et al. [24] proposed a novel algorithm that decomposed the input video into background and residual videos to detect the motion salient regions with much lesser time. In [25], a novel saliency detection algorithm was described, multiscale extrema of local differences measured in the CIELAB space was firstly used to detect potentially salient regions, and then the saliency map was generated by a Gaussian mixture. Chen et al. [26] detailed a new learning framework for video saliency detection, which maked full use of the spatialtemporal consistency to improve the detection accuracy. Instead of direct training on image features, Mai et al. [27] trained a Conditional Random Fields by saliency aggregation on saliency maps. In [28], a regressor based on discriminative regional features is trained and the image saliency is predicted by a random forest model. Recently, convolutional neural network (CNN) techniques had good performance in salience detection [29], and a novel CNN framework integrated with low-level features was proposed to detect salient objects for complex images [30]. Luo et al. [31] designed a simplified CNN with the local and global information, and proposed a loss function to penalize errors on the edge, inspired by the MumfordShah function. A global Recurrent Localization Network (RLN) was proposed to exploit contextual cue of the weighted response map for salient object detection [32].

Point 3: Figure 1 and Figure 2 would be much better to be enlarged or more clearly depicted so as for the readers to easily follow.

Response 3: Thanks. We have made the following modification:

1).

The figure 1 is as follow:

2).

The figure 2 is as follow:

Round 2

Reviewer 1 Report

the manuscript is well revised, the corresponding suggestions have been considered by the authors, therefore, it can be accepted by the Journal of Marine Science and Engineering.

Author Response

Thank you, According to the reviewer's opinions, the English in the article had been comprehensively revised. Such as line 35 "meet"  was modified to "promote"  and so on.